# Oncolytic Virus Therapy Alters the Secretome of Targeted Glioblastoma Cells

**DOI:** 10.3390/cancers13061287

**Published:** 2021-03-14

**Authors:** Jakub Godlewski, Mohamed Farhath, Franz L. Ricklefs, Carmela Passaro, Klaudia Kiel, Hiroshi Nakashima, E. Antonio Chiocca, Agnieszka Bronisz

**Affiliations:** 1Harvey Cushing Neuro-Oncology Laboratories, Department of Neurosurgery, Brigham and Women’s Hospital, Harvard Medical School, Boston, MA 02115, USA; mmohamed@kent.edu (M.F.); f.ricklefs@uke.de (F.L.R.); passarocarmela@gmail.com (C.P.); HNAKASHIMA@BWH.HARVARD.EDU (H.N.); eachiocca@bwh.harvard.edu (E.A.C.); 2NeuroOncology Laboratory, Mossakowski Medical Research Centre, Polish Academy of Sciences, 02-106 Warsaw, Poland; 3Tumor Microenvironment Laboratory, Mossakowski Medical Research Centre, Polish Academy of Sciences, 02-106 Warsaw, Poland; kkiel@imdik.pan.pl

**Keywords:** glioblastoma, oncolytic virus, extracellular vesicles, immune response, secretome, exosomes, cancer

## Abstract

**Simple Summary:**

Proteins secreted by cancer cells in response to oncolytic virus anti-tumor therapy constitute the instructions for the immune cells. Yet as there are hundreds of proteins, including those encapsulated in vesicles, whose message drives the mobilization of immune cells, we aimed to decipher the instruction sent by cancer cells in response to therapy. Searching the cataloged vesicle and vesicle-free secreted proteins, we found that the proteins associated with the favorable survival of brain cancer patients were those that have the power to mobilize the immune cells. Thus, this approach established cancer-secreted contributors to the immune–therapeutic effect of the oncolytic virus.

**Abstract:**

Oncolytic virus (OV) therapy, which is being tested in clinical trials for glioblastoma, targets cancer cells, while triggering immune cells. Yet OV sensitivity varies from patient to patient. As OV therapy is regarded as an anti-tumor vaccine, by making OV-infected cancer cells secrete immunogenic proteins, linking these proteins to transcriptome would provide a measuring tool to predict their sensitivity. A set of six patient-derived glioblastoma cells treated ex-vivo with herpes simplex virus type 1 (HSV1) modeled a clinical setting of OV infection. The cellular transcriptome and secreted proteome (separated into extracellular vesicles (EV) and EV-depleted fractions) were analyzed by gene microarray and mass-spectroscopy, respectively. Data validation and in silico analysis measured and correlated the secretome content with the response to infection and patient survival. Glioblastoma cells reacted to the OV infection in a seemingly dissimilar fashion, but their transcriptomes changed in the same direction. Therefore, the upregulation of transcripts encoding for secreted proteins implies a common thread in the response of cancer cells to infection. Indeed, the OV-driven secretome is linked to the immune response. While these proteins have distinct membership in either EV or EV-depleted fractions, it is their co-secretion that augments the immune response and associates with favorable patient outcomes.

## 1. Introduction

Immunotherapy has had considerable success in treating several types of solid cancers, such as melanoma and lung carcinoma [1]. However, for other tumors, including glioblastoma, immunotherapy has not been as effective [2]. One reason is that glioblastoma is characterized by a highly immunosuppressive “cold” microenvironment that induces systemic immunosuppression [3,4]. As these tumors do not attract and activate immune cells, approaches based on educating immune cells on killing tumor cells, utilized in “hot” immuno-activating cancers, have not been successful in brain tumor clinical trials. In this context, the use of immune-stimulatory approaches, like therapy with oncolytic viruses (OV), is promising [5].

OV therapies combine cytolytic antitumor effects with inducing an immune response against infected, and possibly even non-infected, cancer cells by the “bystander effect” [6]. Yet, long-term survival with OV therapy remains uncommon [4,7]. OVs based on herpes simplex virus type 1 (HSV1) have seen widespread application in cancer clinical trials, with one OV (Imlygic) currently approved for melanoma [8]. rQNestin34.5 is an oncolytic HSV tested in phase I clinical trial for glioblastoma [9,10], which we used to infect patient-derived glioblastoma stem-like cells (GSCs), exhibiting diverse phenotypes, transcriptomic subtype identities [7,11,12], and significantly variable response to OV [13,14].

The subset of proteins expressed and secreted into the extracellular space encompasses 13–20% of the proteome. It includes cytokines, growth factors, extracellular matrix proteins, and shed receptors, which mediate information between cells [15,16,17]. In solid tumors, the cancer cell secretome can help to evade immune system surveillance, creating a tumor-friendly environment [18,19]. Engineered OVs, often applied along with chemotherapy [20,21], are designed to infect and lyse cancer cells and, after successful replication, to release new infectious virus particles to infect the remaining tumor [22]. Studies have shown that EV are swiftly released upon OV treatment at the early stage of the infection when cell viability has not yet been compromised, suggesting that OV can transmit infection messaging via EV signaling even before lytic viral release [21].

In glioblastoma, OV infection alone may stimulate host antitumor immune system responses [23,24,25] by triggering the “immuno-secretion” that releases cancer antigens and cytokines to stimulate an immune response, which can identify and eliminate tumor cells. Identifying the secreted factors that can be used as biomarkers of infection rate or therapeutic co-targets, would help develop appropriate testing and improve therapeutic efficiency.

Numerous human genes (39%, [26]) encode for proteins that are predicted to have either a signal peptide or at least one transmembrane region suggesting active secretion, or positioned within one of the numerous membranous systems in the cell. The broad term “secretion” encompasses both vesicle-depleted protein, as well as proteins encapsulated within vesicles, including microvesicles (100–1000 nm in diameter) shed from the plasma membrane, and exosomes (30–150 nm in diameter) released via endosomal-exocytosis events [27]. Factors present in both these organelles, jointly named EV, account for almost half of the collective secretome [28]. Thus, the secretome can be recognized as a sum of two roughly equal EV and EV-depleted fractions.

The main problem, which prompted us to perform this study, was that the analysis of secretome analysis within a tumor microenvironment is highly difficult, if not impossible. Meanwhile, transcriptome analysis is feasible, affordable, and fast. This problem can be addressed by the association between secretome proteins and their respective transcripts.

Testing the hypothesis that OV infection remodels cancer cell secretome composition toward an immune-responsive mode, we included additional dimensions by dividing the cell transcriptome into EV and EV-depleted secretomes. We used GSCs to understand the transcriptional heterogeneity of glioblastoma in response to OV infection. Despite vast differences in the infectivity rate, OV infection forced GSC to accumulate transcripts encoding for secretome proteins, and the OV-dependent transcriptome signature was correlated with a secretome profile. While both EV and EV-depleted subsets of secreted proteins had distinct and not overlapping membership, they showed an association with antitumor immune system response, which was blueprinted in the glioblastoma patients’ survival outcome [29]. Thus, our work provides a roadmap of the glioblastoma secretome programs and their plasticity in response to OV infection.

## 2. Materials and Methods

### 2.1. Cell Culture

GSCs were cultured as neurospheres under stem cell-enriching conditions, using neurobasal medium supplemented with 1% glutamine, 2% B27, and 20 ng/mL EGF and FGF2 (epidermal growth factor and fibroblast growth factor 2, respectively) using ultra-low attachment plates/flasks. The unique identity of cultured patient-derived cells was confirmed by short tandem repeat analysis [30]. All GSCs used in this study were isocitrate dehydrogenase (IDH) wild type [11]. Mycoplasma testing was routinely performed by PCR.

### 2.2. Viral Particle Preparation

rQNestin34.5, a virus used in this study has been reported previously [9,10]. In brief, the rQNestin34.5 recombinant virus contains a deletion of the *UL39* gene (encoding ICP6) and both copies of the γ_1_34.5 genes, with the addition of one copy of the γ_1_34.5 gene under the control of nestin promoter inserted into the deleted UL39 locus. K26GFP virus was obtained from P. Desai (Johns Hopkins University School of Medicine, Baltimore, MD, USA) [31]. Viral stocks were prepared as previously described using the 7b cell line [32] in 5- or 10-layer Nunc Cell Factories (Thermo Fisher Scientific, Waltham, MA, USA) at an MOI of 0.005 to 0.01, at 37 °C in serum-free media (Thermo Fisher Scientific). Briefly, after the 2-h adsorption period, media with fetal bovine serum (Sigma-Aldrich, St. Louis, MO, USA) were added to a final concentration of 2%. At 24 h after infection, cultures were shifted to 33 °C and checked daily for the appearance of cytopathic effects (CPE). When cultures reached approximately 80% CPE, the cultures were exposed to an additional 0.45 M of NaCl (Sigma-Aldrich) and 100 μg/mL dextran sulfate (Sigma-Aldrich). Twenty-four hours later, the cultures were spun at 660× *g* in a table-top refrigerated centrifuge (Thermo-Fisher), and the supernatants were purified under GLP-like conditions. The virus was resuspended in D-PBS (Sigma-Aldrich), and sterile glycerol (Sigma-Aldrich) was added to a final concentration of 10%, aliquoted into cryovials (Corning), and stored at −80 °C. Titers were determined in triplicate in 7b cells according to standard protocols, and the number of viral genomes was determined by qPCR, as previously described [33].

### 2.3. Viral Infection Assay

GSCs were plated at 250,000 cells per 6-well plate, and after 2 h, the 1 mL of Neurobasal medium containing the diluted virus stock at 37 °C was added at MOI 0.1. After 12 h, luciferase assays were performed using a Luciferase Assay Kit (Promega, Madison, WI, USA), and using a microplate reader (POLARStar Omega, BMG Labtech, Cary, NC, USA). Bioluminescence measured as relative luciferase units (RLU) indicated HSV-1-mediated transgene expression.

Secretome EV/EV-depleted fractionation. GSC conditioned medium was collected from cells after 12 h after OV infection, centrifuged at 300× *g* for 10 min, 2000× *g* for 10 min at 4 °C, and filtered through a 0.22 μm filter (Millipore, Billerica, MA, USA), and then ultra-centrifugated at 100,000× *g* for 90 min at 4 °C. Supernatants were decanted and used as EV-depleted fraction, while EV pellets (EV + HSV fraction) resuspended in PBS were collected by a second ultra-centrifugation at 100,000× *g* for 90 min at 4 °C and used as EV fraction, respectively.

### 2.4. Quantitative PCR

Total RNA was extracted using a standard Trizol protocol (Invitrogen). The RNA quantity and quality were measured using a NanoDrop 2000 (Thermo Scientific). For mRNA analysis, 3 μg total RNA was treated with DNase (Promega) for 2 h to remove genomic DNA. Reverse transcription (RT) was performed using random hexamers and iScript (BioRad), and quantitative PCR (qPCR) was performed using TaqMan or SYBR Green master mix (Applied Biosystems, Foster City, CA, USA). Amplification was performed using the StepOnePlus Real-Time PCR System (Applied Biosystems, Foster City, CA, USA), and the software determined Ct thresholds. Expression was quantified using the ΔΔCt method, using 18S rRNA as a reference gene.

### 2.5. Immunofluorescence

Paraformaldehyde fixed (4%), permeabilized with 0.1% Triton™ X-100 for 15 min, cells were blocked using 10% normal rabbit serum (Jackson ImmunoResearch Laboratories, Inc., West Grove, PA, USA). Incubation with primary antibodies: anti-TPM3 in concentration 3 µg/mL, MST1 (10 µg/mL), CD276 (1 µg/mL), CD320 (1 µg/mL) diluted in 0.1% BSA, was performed overnight at 4 °C. Afterward, slides were washed in TBS buffer (50 mM Tris-HCl, pH 7.4; 150 mM NaCl) and incubated for one hour with secondary antibodies with Alexa Fluor 594 (0.4 µg/mL, Thermo Scientific). Nuclei were stained with ProLong™ Diamond Antifade Mountant with DAPI (Thermo Fisher). The images were captured at 60X magnification using a confocal microscope, Zeiss LSM710.

### 2.6. Electron Microscopy

Cell or EV pellets were collected and washed three times with PBS. EV-depleted samples were transferred to a 3 kDa molecular weight cut-off device (Millipore, Billerica, MA, USA) and centrifuged at 13,000× *g* for 30 min to concentrate the protein. Samples were then fixed in 2.5% glutaraldehyde (catalog 16220, Electron Microscopy Sciences, Hatfield, PA, USA) in cacodylate buffer for 4 h at 4 °C. Cells were then washed three times with cacodylate buffer followed by post-fixation in 1% OSO4 (catalog 0972A, Polysciences, Warrington, PA, USA) in cacodylate buffer for 1 h at 4 °C. After three cacodylate buffer washes, the pellets were then dehydrated through graded alcohol, embedded in Epon/Araldite, and polymerized at 60 °C overnight. Blocks were then sectioned at 70 to 80 nm and collected on the grids. The grids were first stained with 1% aqueous uranyl acetate (catalog NC0740462, Fisher Scientific) for 12 min, followed by a ddH_2_O rinse. The grids were then stained with Reynolds lead citrate (catalog 17900, Electron Microscopy Sciences) for 5 min in a moisture-free chamber. After the rinse with ddH_2_O, the grids were set to dry before analysis via Hitachi H-7650 SEM/TEM Hitachi S-4800. For the analysis of CD63, samples were blocked with 1% BSA in PBS for 1 h at room temperature and then with primary antibody against CD63 (10 µg/mL) overnight at 4 °C. After that samples were washed three times with PBS and then incubated with secondary antibody for 1 h at room temperature. After washing, images were captured.

### 2.7. Gene Microarray

Transcriptome expression analysis was performed on total RNA extracted from GSCs (*n* = 12) mock or OV treated. Array Star Inc. performed RNA labeling and array hybridization. Briefly, total RNA from each sample was linearly amplified and labeled with Cy3-UTP. The labeled antisense RNAs (cRNAs) were purified using an RNAeasy mini kit (Qiagen, Germantown, MD, USA). The labeled cRNAs’ concentration and specific activity (pmol Cy3 per μg of cRNA) were measured by a NanoDrop ND-1000. The labeled cRNA (1 μg each) was fragmented by adding 11 μL 10× blocking agent and 2.2 μL 25× fragmentation buffer, then heated at 60 °C for 30 min, and finally, 55 μL 2× GE hybridization buffer was added to dilute the labeled cRNA. The hybridization solution (100 μL) was dispensed into the gasket slide and assembled to the gene-expression microarray slide. The slides were incubated for 17 h at 65 °C in an Agilent hybridization oven. Agilent Feature Extraction software (version 11.0.1.1) was used to analyze the acquired array images. Quantile normalization and subsequent data processing were performed using GeneSpring GX v12.1 software (Agilent Technologies Santa Clara, CA, USA).

### 2.8. Arrays Data Analysis

After quantile normalization of the raw data, genes that had flags in at least 3 out of 12 samples as detected (“All Targets” “Value”) were chosen for further data analysis. The raw expression intensities were log2 transformed and normalized by quantile normalization. Differential analysis between two groups was performed by t-test. The cutoffs were *p* ≤ 0.05 and fold change ≥2.0. Normality was assumed for log2 transformed normalized intensity values across samples per gene, >90% of genes in our dataset passed the Shapiro–Wilk normality test. Differentially expressed transcripts with statistical significance were identified through volcano plot filtering (GraphPad Prism, San Diego, CA, USA). Hierarchical clustering was performed using dchip software. 

### 2.9. Mass Spectrometry-Based Proteomics

Global proteomics profiling analysis was performed on EV, and EV-depleted fraction from GSCs mock or OV treated (*n* = 6 per group), as we described previously [34,35]. All mass spectra were acquired at the Bioproximity LLC using Standard Label-Free Quantitative Proteome Profiling Assays Top 4000. Samples were prepared for digestion using the filter-assisted sample preparation method [36]. Briefly, the samples were suspended in 8 M urea, 50 mM Tris-HCl, pH 7.6, 3 mM dithiothreitol, sonicated briefly and incubated in a Thermo-Mixer at 40 °C, 1000 rpm for 20 min. The remaining sample was buffer exchanged with 8 M urea, 100 mM Tris-HCl, pH 7.6, then alkylated with 15 mM iodoacetamide. The urea concentration was reduced to 2 M. Samples were digested using trypsin at an enzyme to substrate ratio of 1:100, overnight, at 37 °C in a Thermo-Mixer at 1000 rpm. Digested peptides were collected by centrifugation and desalted using C18 stop-and-go extraction (STAGE) tips [37]. Briefly, for each sample, a C18 STAGE tip was activated with methanol, then conditioned with 60% acetonitrile, 0.5% acetic acid, followed by 2% acetonitrile, and 0.5% acetic acid. Samples were loaded onto the tips and desalted with 0.5% acetic acid. Peptides were eluted with 60% acetonitrile, 0.5% acetic acid, and lyophilized in a speed vac (Thermo Fisher Scientific) to near dryness, ~2 h. For LC-MS/MS, each digestion mixture was analyzed by UHPLC–MS/MS. LC analysis was performed on an Easy-nLC 1000 UHPLC system (Thermo). Mobile phase A was 97.5% MilliQ water, 2% acetonitrile, and 0.5% acetic acid. Mobile phase B was 99.5% acetonitrile and 0.5% acetic acid. The 240 min LC gradient ran from 0% B to 35% B over 210 min, then to 80% B for the remaining 30 min. Samples were loaded directly into the column. The column was 50 cm × 75 μm I.D. and packed with two μm C18 media (Thermo Fisher Scientific). The LC analysis was interfaced to a quadrupole-Orbitrap mass spectrometer (Q-Exactive, Thermo Fisher) via nanoelectrospray ionization using a source with an integrated column heater (Thermo Fisher Scientific). The column was heated to 50 °C. An electrospray voltage of 2.2 kV was applied. The mass spectrometer was programmed to acquire, by data-dependent acquisition, MS/MS from the top 20 ions in the full scan from 400 to 1200 *m*/*z*. Dynamic exclusion was set to 15 s, singly-charged ions were excluded, isolation width was set to 1.6 Da, full MS resolution to 70,000, and MS/MS resolution to 17,500. The normalized collision energy was set to 25, automatic gain control to 2e5, max fill MS to 20 ms, max fill MS/MS to 60 ms, and the underfill ratio to 0.1%.

### 2.10. Proteomic Data Analysis

Mass spectrometer RAW data files were converted to MGF format using the msconvert tool [38]. Briefly, all searches required ten ppm precursor mass tolerance, 0.02 Da fragment mass tolerance, strict tryptic cleavage, 0 or 1 missed cleavages, fixed modification of cysteine alkylation, variable modification of methionine oxidation, and expectation value scores of 0.01 or lower. MGF files were searched using the human and viral sequence library. MGF files were searched using X!! Tandem [39] using both the native [40] and k-score [41] scoring algorithms, and by OMSSA [42]. All searches were performed on Amazon Web Services-based cluster compute instances using the Proteome Cluster interface. XML output files were parsed, and non-redundant protein sets were determined using Proteome Cluster [43]. MS1-based features were detected, and peptide peak areas were calculated using OpenMS [44]. Proteins were required to have one or more unique peptides across the analyzed samples with *E*-value scores of 0.01 or less. For differential proteome analysis, we used the PepHits value to calculate the number of identification events to peptides mapping to this protein identifier. As peptides may be identified multiple times, the number of identifications correlates with protein abundance.

TCGA data analysis. The collection of the data from TCGA (The Cancer Genome Atlas Research, 2008) for glioblastoma was compliant with all applicable laws, regulations, and policies for the protection of human subjects, and necessary ethical approvals were obtained. Experimental and clinical data were downloaded (https://tcga-data.nci.nih.gov/ accessed on: 1 March 2020), as described in The Cancer Genome Atlas Research Network.

Displaying a summary of experimental data associated with selected genes: The samples (columns on the heatmap) were annotated in two ways: first, according to cluster membership (the optimal number of clusters was determined using NbClust); second, by inspecting the status of a prognostic index (which was computed by weight, averaging the gene expressions with the regression coefficients of a multi-gene Cox proportional hazards model). The gene names were annotated with their respective hazard ratios in a multi-gene Cox proportional hazards model. When search results involve more than 50 genes, we filtered them by keeping the 50 genes whose expression was the most varied among the samples.

### 2.11. Performing Gene Survival Analysis

The Kaplan–Meier survival curve compares samples stratified according to gene expression levels. The default options stratified samples into two groups: those with expression levels smaller than the median over the subgroup, and those with expression levels higher than the median. For gene searches that resulted in multiple hits, we analyzed how the expression profiles impacted survival. We performed two types of survival analyses: first, the optimum clusters were selected by stratifying the samples according to the heatmap cluster membership (see the first annotation bar), where the optimal number of clusters was picked out algorithmically using silhouette width index. Next, we used a Kaplan–Meier model to analyze the differences in survival between groups using a log-rank statistic. These analyses were performed using the “NbClust” package in R.

Classify tumor samples based on mRNA expression profiles. SVM data classification and function approximation, introduced by Vapnik [45], is a binary classifier (trained on a set of labeled patterns called training samples) [46]. This SVM strategy was used to investigate the possibility of identifying different prognostic subsets of patients based on their clinicopathologic features and immunomarkers [47]. We used this approach to classify cell transcriptome into a molecular subtype of glioblastoma and immune genes enrichment between mock and OV treated cells. The subtype call used here by SVM is done using a linear kernel with 10-fold cross-validation, using the ksvm function of the kernlab package. As a training dataset for the glioblastoma, the TCGA glioblastoma samples described by Wang and colleagues were used [48]. 

### 2.12. Quantification and Statistical Analysis

Graphs (scatters plot, box plots, PCA) were generated, and statistical analyses were performed using GraphPad Prism 7. Statistical parameters, including the value of n, statistical test, and statistical significance (*p*-value), are reported in the figures and their legends. For studies involving cell culture, replicates refer to technical or biological replicates. No statistical methods were used to predetermine the sample size. Statistical tests were selected based on the desired comparison. Paired two-tailed t-tests were used to assess significance when comparing data between biological variances (mock vs. OV), while unpaired for technical replicates One-way ANOVA was used to determine significance when comparing data between ≥3 variances; significant ANOVA results were followed by post hoc testing either comparing every mean with every other mean (Tukey’s multiple-comparison test), or comparing every mean to the control mean (Dunnett’s multiple-comparison test).

### 2.13. Key Resources Table 

All information about the reagents used in the experiments, primers for the PCR reaction, and the software used in this publication are summarized in Appendix A [10,29,49,50,51,52,53,54,55].

## 3. Results

### 3.1. OV Infection Alters the Transcriptome of Patient-Derived GSCs toward the Extracellular Mode

We initially asked whether OV infection changed the transcriptome of a panel of GSCs. Glioblastomas and GSCs display diverse transcriptome subtypes, and we have previously shown that we can model this heterogeneity in both in vitro and in vivo models [56,57,58]. We thus selected six GSCs with diverse transcriptomes (Appendix A). We noticed a dualism in response to OV infection among these different GSCs. Twelve hours after infection with an OV that expresses luciferase, GSCs exhibited either high or low bioluminescence, indicating differential infection/replication of the OV (Figure 1a). To determine what changes in the transcriptome accompanied such differential infection, we carried out a gene array that indeed did show a significant shift of transcriptomic signatures (Figure 1b and Appendix A). Overall, for all six GSCs, there were 704 transcripts up-regulated, and only 350 downregulated, suggesting an active response to the OV challenge (Figure 1c).

Interestingly, both GSC subgroups showed similar transcriptome changes, regardless of their sensitivity to OV infection, as they clustered together after infection instead of clustering with their parental, mock-infected cells. Still, the magnitude of transcriptome rearrangements (i.e., number of affected transcripts) was more profound in the GSCs that were more easily infected by OV (increased sensitivity GSCs) than the ones that showed reduced sensitivity (Figure 1d and Appendix A). Transcripts up-regulated upon OV infection encoded a multitude of proteins linked to membranes, vesicles, and extracellular components and functionally connected to gene expression and RNA metabolism. Conversely, transcripts downregulated in OV-infected GSCs were linked to intracellular proteins with intrinsic metabolic functions (Figure 1e and Appendix A).

These findings showed that the transcriptome of all tested GSCs changed significantly, regardless of their sensitivity to the initial OV infection. These transcriptomes clustered similarly after OV, regardless of their pre-infection gene expression signature. Significantly, most up-regulated transcripts encoded for proteins annotated as membrane-bound, vesicle-bound, or extracellular, suggesting that the composition of the secretome may be one of the most profound changes in GSCs in response to OV exposure.

### 3.2. OV Infection Leads to Quantitative and Qualitative Changes in Both the EV-Depleted and EV Secretome of GSCs

We next analyzed both the quantitative and qualitative changes in the GSC secretome after OV infection (flow chart depicting experimental procedures, see Appendix A). First, we wanted to determine if the increase in transcripts related to the membrane and vesicle dynamics 12 h after OV infection was due to cytolysis. We utilized an OV that expresses GFP on its capsid [31] and glioma cells that express palm tdTomato localized to membranes. This allowed us to monitor cells in real-time after infection. As cell membranes were still intact after 12 h (Figure 2a), we selected this time point as optimal for the analysis of the secretome. This also suggested that the increase in transcripts related to cellular membranes was due to active secretion and not disintegrating cellular membranes. This and the electron microscopy imaging (Appendix A), allowed us also to detect OVs in proximity with EVs, thus indicating that human and viral molecules can be expected to intermingle within the same compartment. Next, to gain more insight into the secretome composition, we separately analyzed its two fractions: EV-depleted and encapsulated within or bound to EVs, with the latter fraction expected to contain HSV particles. While highly variable, the enriched CD63 protein was associated exclusively with the EV fraction, and did not contaminate the EV-depleted fraction. (Figure 2b,c). OV infection significantly enhanced the secretion of EV-depleted proteins in the population of GSCs with increased OV sensitivity, while there was much less of an increase in those with reduced OV sensitivity (Figure 2d,e and Appendix A). While there were no significant changes in the frequency and size of secreted EVs (Figure 2f,g and Appendix A) or total protein content (Appendix A) between both groups, the enrichment of particles >220 nm after OV infection, may be indicative of the presence of viral particles. Next, we used a differential proteome analysis to test the co-presence of protein markers in both fractions that revealed a fraction-dependent distribution of selected markers including CD63 (Appendix A). While the enrichment of human proteins in EVs was not significantly changed in the GSCs with reduced OV sensitivity, it showed somewhat diminished levels in the GSCs with increased OV sensitivity. Yet, as it was substituted by viral proteins, the overall total (human and viral) protein amount was changed significantly (Figure 2h,i).

These results thus demonstrated that GSCs responded to OV infection by secretome rearrangements. First, we documented that with cells still intact, any observed change was due to active response in secretion leading to quantitative and qualitative changes in the secretome, not by cell lysis and disintegration. The observed significant quantitative increase in the EV-depleted fraction of secretome was accompanied by substantial qualitative changes in the EV protein content (but not overall EV secretion), while the increase of viral protein content substituted a decrease in human proteins. These changes suggested dynamic rearrangements in both fractions of the secretome upon OV infection, and raised the question of the nature of and consequences of these alterations.

### 3.3. Proteins Secreted by Mock- and OV-Infected GSCs End Up in Non-Overlapping EV-Depleted and EV Fractions

These discoveries prompted us to take a closer look at the composition of both fractions of the secretome. We performed mass spectroscopy data analysis of these fractions (Appendix A), and we found that the EV-depleted proteome was highly dissimilar from the EV proteome, without overlapping proteins belonging to both fractions (Figure 3a). Functionally, both proteomes were annotated to different pathways. Although proteins from both fractions ascribed predominantly to secretory processes, EV-depleted proteins were overwhelmingly implicated in intercellular communication processes, such as cell surface receptor, receptor–ligand activity, etc. Still, EV proteins were engaged in intracellular functions, such as development and metabolism (Figure 3b and Appendix A). We then analyzed changes after infection separately: a significant shift in the proteome composition was more pronounced in the EV fraction and among GSCs with increased OV sensitivity (Figure 3c–f). We selected one protein from each category: EV-depleted down-, and up-regulated (TPM3 and MST1, respectively) and EV down-, and up-regulated (CD276 and CD320, respectively), as a proof-of-concept for further analysis and validation. We confirmed the mass-spectroscopy results for both down- and up-regulation of these proteins (Figure 3g,h and Appendix A). To strengthen these results, we also included the analysis of their subcellular localization in cancer cells other than glioblastoma, which validated the predicted fraction membership (Appendix A). Finally, we demonstrated that protein content changes were regulated transcriptionally as the proteins’ levels closely mirrored their corresponding mRNAs (Figure 3i,j).

These results thus showed that the proteome composition of the EV-depleted and EV fractions of the secretome of GSCs was different, annotating largely to intercellular signaling for EV-depleted secretome, and mostly intracellular signaling for EV secretome. OV infection caused a significant, transcriptionally-enforced shift in the composition of both fractions.

### 3.4. The Re-Configuration of GSCs’ Transcriptome upon OV Infection Predicts the Composition of Their “Immune Mode” Secretome

The previous findings prompted us to scrutinize further the relationship between secretome proteins and their respective genes. As all GSCs (regardless of sensitivity to infection) remodeled transcriptome similarly, with only the magnitude of response differed, we narrowed down the list of proteins that were significantly deregulated by OV treatment in both fractions (EV and EV-depleted) to 1919 entries (Appendix A). There was sufficient power in this analysis to cluster GSCs genes into mock and OV clusters (Appendix A). Importantly as many as 78% (1500 out of 1919) showed a significant correlation between mRNA and protein (Figure 4a,b). This approach allowed the identification of 524 and 1030 protein/mRNA tandems that correlated in the EV-depleted and EV fractions, respectively (Figure 4c,d). We next aimed to determine if these genes were linked to the survival of glioblastoma patients. According to their survival and prognostic index, both EV-depleted and EV secretome protein encoding genes stratified patients to better and poorer outcomes; furthermore, predictive genes were found among genes that were both down- and up-regulated upon OV infection (Figure 4e,f and Appendix A). Importantly, all these predictive genes belonged to the same broad functional category–immune system/immune response (Appendix A).

We deconvoluted the “immune mode” in all genes encoding for secretome proteins significantly affected by OV treatment to explore this further. To this end, we demonstrated that GSCs treated with OV significantly enriched components required for stromal communication, and boosted expression of immune response genes encoding for their secretome in various immune cell-type-specific profiles (Figure 4g and Appendix A). Among those especially prominent were genes engaged in T-cell and B-cell activity. Finally, among these 1500 genes, we selected 220 genes categorized as the “immune mode” cluster [59]. Using this platform, we demonstrated that both EV-depleted and EV fractions of the secretome are engaged in immune system/immune response activity, which encompasses both innate and adaptive immunity (Figure 4h,i and Appendix A).

These findings supported the hypothesis that changes in cells’ transcriptome control the GSC response to OV infection, and that this gets reflected in the secretome’s composition. Moreover, this transcriptome signature is clinically relevant, as it is predictive of glioblastoma patients’ survival. Finally, we demonstrated that affected proteome/transcriptome tandems were thoroughly connected to various aspects of immune system/immune response, suggesting that, indeed, OV infection significantly affects GSCs immunogenicity, thus supporting the overarching hypothesis of the value of OV as an anti-cancer vaccine. Of note, the induction of both T-cell- and B-cell-related genes points toward increased immune cell-mediated cytotoxicity and boosted immune response memory.

## 4. Discussion

During the last decade or so, oncolytic viruses’ therapeutic usage has evolved from primarily antitumor cytotoxicity, toward enhancing the immune response [6,60,61]. The engineering of the oncolytic virus, and clinical trials [62] with oncolytic virus immunotherapy have provided early-stage results showing that some tumors respond by changing the “cold” tumor immune environment toward a “hot” one in response to the oncolytic virus [63,64]. As glioblastoma is a phenotypically and transcriptionally heterogeneous disease, it requires identifying and addressing areas where the treatment is disadvantaged by such high heterogeneity. However, by looking into OV-dependent transcriptional changes in diverse patient-derived cells, we ultimately found cell type-independent responses in genes encoding for proteins annotated for secretome.

Glioblastomas are mostly resistant to immune system surveillance and actively suppress immune response via the release of tumor cells’ secretomes. This includes proteins encapsulated within EVs that contain surface proteins engaged in the communication with immune cells [65,66,67,68,69,70,71,72]. Knowledge of EV and EV-depleted secretome rearrangements driven by OV expanded the concept of cancer immunosuppressive function beyond the boundary of tumor cells [49,73,74,75]. That is important, as the secretome may act as a niche-dependent factor affecting cells in neighboring anatomic features, and because co-targeting of the secretome players would benefit immunotherapy [48,76,77,78,79,80].

The origin of glioblastoma as a result of virus infection was proposed, and especially promising data were found for cytomegalovirus [81]. However, low cellular detection caused some skepticism in the community. While not directly indicated by the current analysis, it will be interesting to find whether the deposition of cytomegalovirus proteins is detectable only in glioblastoma-derived EVs. This would support our observation that while viral microRNAs are not detectable in glioblastoma cells, we found them in EVs [57]. The explanation could be as trivial as a different ratio of viral vs. cell host proteome, yet the overarching explanation is still lacking, and will be the subject of follow-up studies.

Using our model of GSC secretome response to OV treatment, we provided detailed insight into the complexity of glioblastoma immune response by demonstrating the following characteristics: (i) OV infection affects transcripts encoding for secretome proteins; (ii) while OV infection does not affect EV secretion, it rearranges EV content; (iii) EV and EV-depleted secretome proteins have distinct membership; and iv) GSC secretome “immune mode” response contributes to the therapeutic effect of OV by enhancing their immunogenicity.

There have been several clinical studies using oncolytic viruses against glioblastoma [82,83]. Yet, none of these studies have shown the importance of GSC secretome in response to the treatment. Our discovery was that the GSC secretome proteins (divided into EV and EV-depleted fractions) associated with changes in transcript levels in response to OV infection are linked to both T-cell-mediated cytotoxicity and B-cell-dependent immune response memory. This sheds new light on oncolytic virus therapy against glioblastoma, and points toward using an OV/EV hybrid as an anti-tumor remission vaccine.

## 5. Conclusions

Glioblastoma cells respond to the OV infection by boosting their secretome, both EV and EV-depleted fractions, and funneling it toward immune response mode, thus increasing its immunogenicity. We thus argue that OV-based therapies should be regarded as anti-cancer vaccines. 

## Figures and Tables

**Figure 1 cancers-13-01287-f001:**
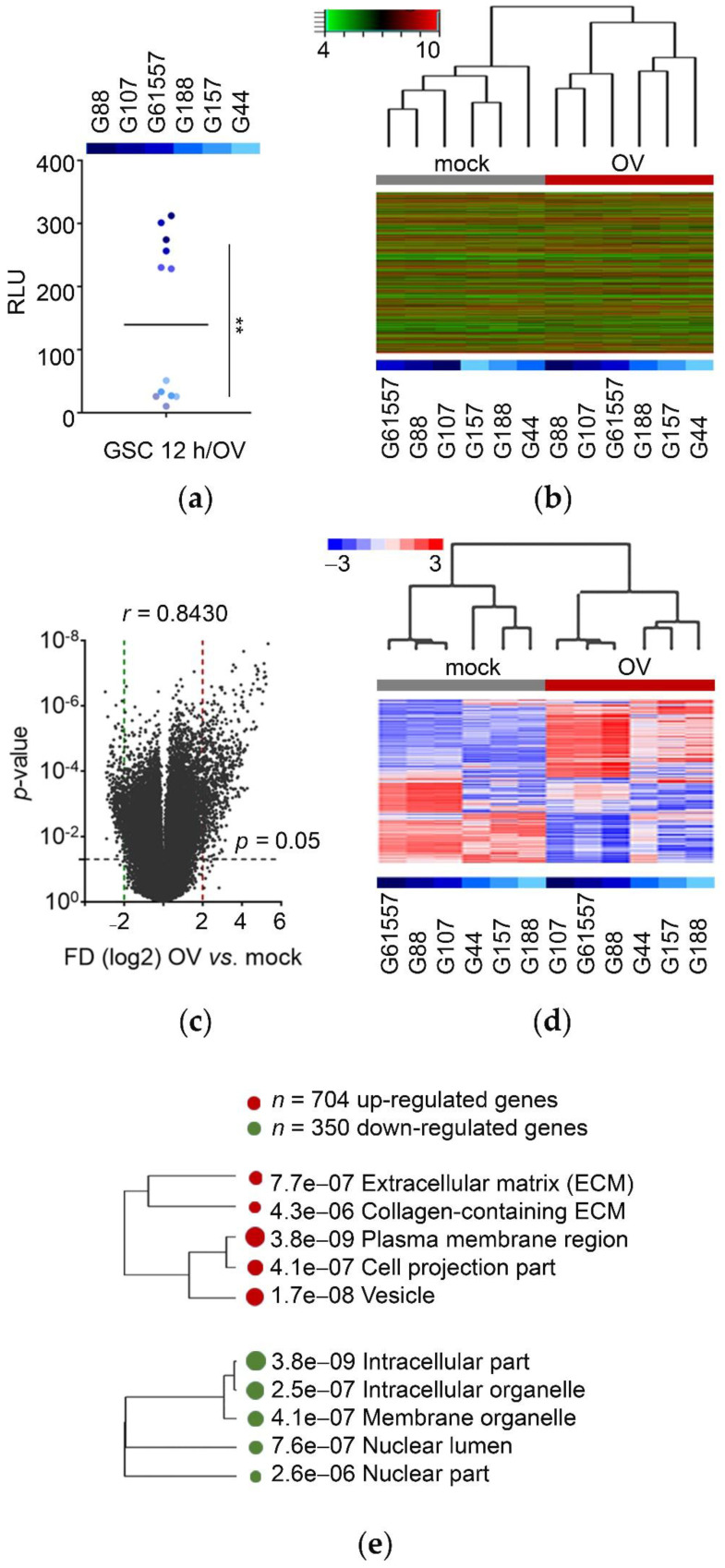
Oncolytic virus (OV) infection alters the transcriptome of patient-derived glioblastoma stem-like cells (GSCs) toward extracellular mode. (**a**) GSCs displayed variable sensitivity to OV infection. GSCs (*n* = 6, in duplicates, see color bar above) were mock or OV infected and harvested after 12 h, followed by bioluminescence assay (measured as RLU). Scatter plot with mean, ** *p*-value < 0.0001. (**b**) OV infection shifts the transcriptome signature of GSCs. Heatmap with unsupervised hierarchical clustering for GSCs mock or OV infected (*n* = 6 per variant) based on mRNA transcripts (*n* = 31,555 genes on Arraystar™ human mRNA array). (**c**) GSC transcriptome is upregulated upon OV infection. Volcano plot for GSCs mock or OV infected (*n* = 6 per variant); dashed lines indicating *p*-value and fold difference (FD) cutoffs (*n* = 1054 genes on Arraystar™ human mRNA array (704 upregulated and 350 downregulated), FD > 2, *p*-value < 0.05). (**d**) The magnitude of GSC transcriptome rearrangements depends on OV-sensitivity. Heatmap with supervised hierarchical clustering for GSCs mock or OV infected (*n* = 6 per variant) based on mRNA transcripts (number of transcripts indicated on panel c). (**e**) Transcripts affected by OV encode for proteins from different cellular and extracellular compartments. A hierarchical clustering tree summarizing the subcellular localization of genes affected by OV infection based on subcellular compartments; up-regulated (red, *n* = 704) and downregulated (green, *n* = 350). Bigger dots indicate more significant *p*-values (number of transcripts indicated on panel **c**).

**Figure 2 cancers-13-01287-f002:**
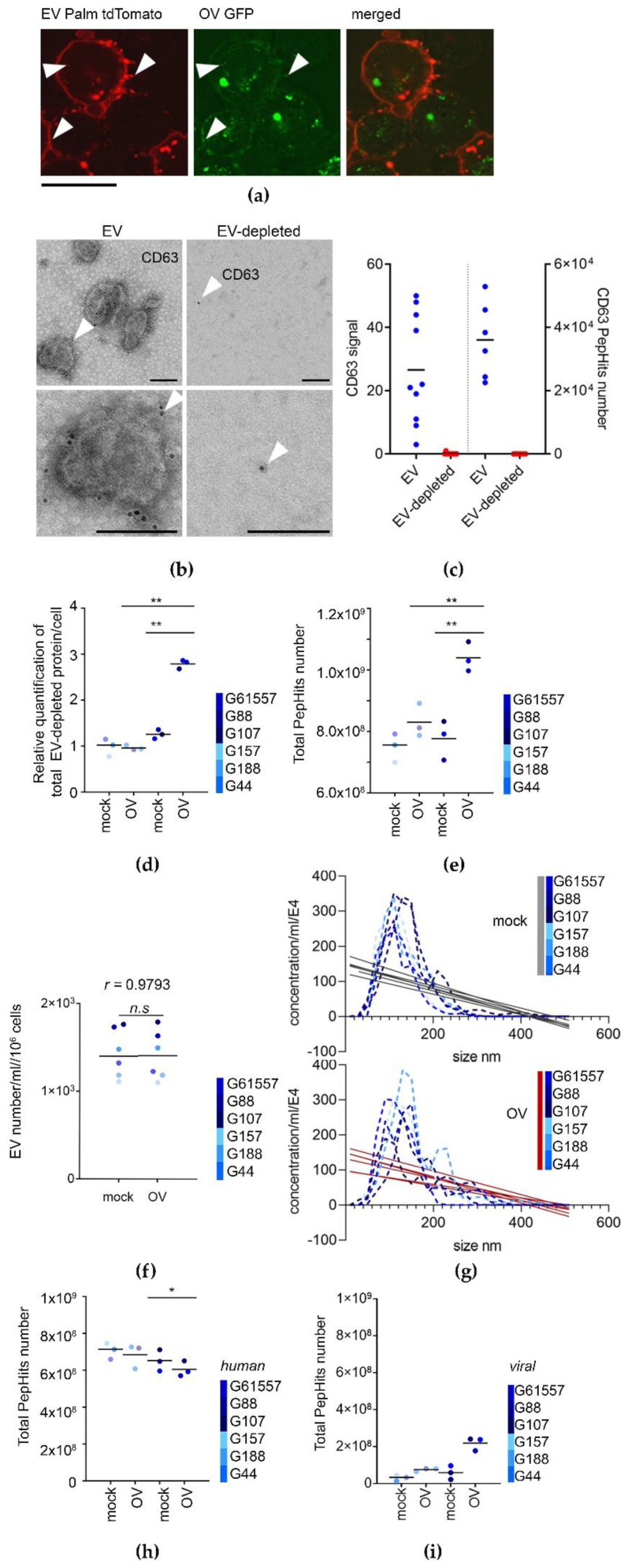
OV infection leads to quantitative and qualitative changes in both the extracellular vesicles (EV)-depleted and EV secretome of GSCs. (**a**) OV co-localizes with EVs within GSCs and upon secretion. Representative micrographs of cells 12 h after infection and colocalization of OV/EV by fluorescent confocal microscopy. Arrows point the co-localization signal; scale bar: 25 µm; red signal: palmitoylated tdTomato GSCs/EVs, green signal: GFP OV. (**b**,**c**) CD63 was enriched in EV but not in the EV-depleted fraction. Representative micrographs of immunodetection of CD63 by transmission electron microscopy; scale bar: 500 nm (**b**). Scatter plot analysis of CD63 staining (signal per random field, *n* = 10 per variant) and mass spectroscopy (intensity of peptide hits, *n* = 6 per variant) (**c**). (**d**,**e**) OV infection enhances EV-depleted protein release in the subpopulation of GSCs. Mass spectroscopy analysis of EV-depleted protein secreted by mock or OV infected GSCs by relative quantification of total protein content per cell (**d**, *n* = 3 per variant) and total peptide hits number (**e**), *n* = 3 per variant), ** *p*-value < 0.01. (**f**,**g**) OV infection does not affect EV number and size. Scatter and distribution plots for EV number (**f**) and size (**g**) secreted by mock or OV infected GSCs (*n* = 6 per variant) by NanoSight™ profiles. (**h**,**i**) GSC EVs carry both human and viral proteins. Mass spectroscopy analysis of EV protein secreted by mock or OV infected GSCs by total peptide hits number. Data for human (**h**) and viral (**i**) proteins separately (*n* = 3 per variant), * *p*-value < 0.05.

**Figure 3 cancers-13-01287-f003:**
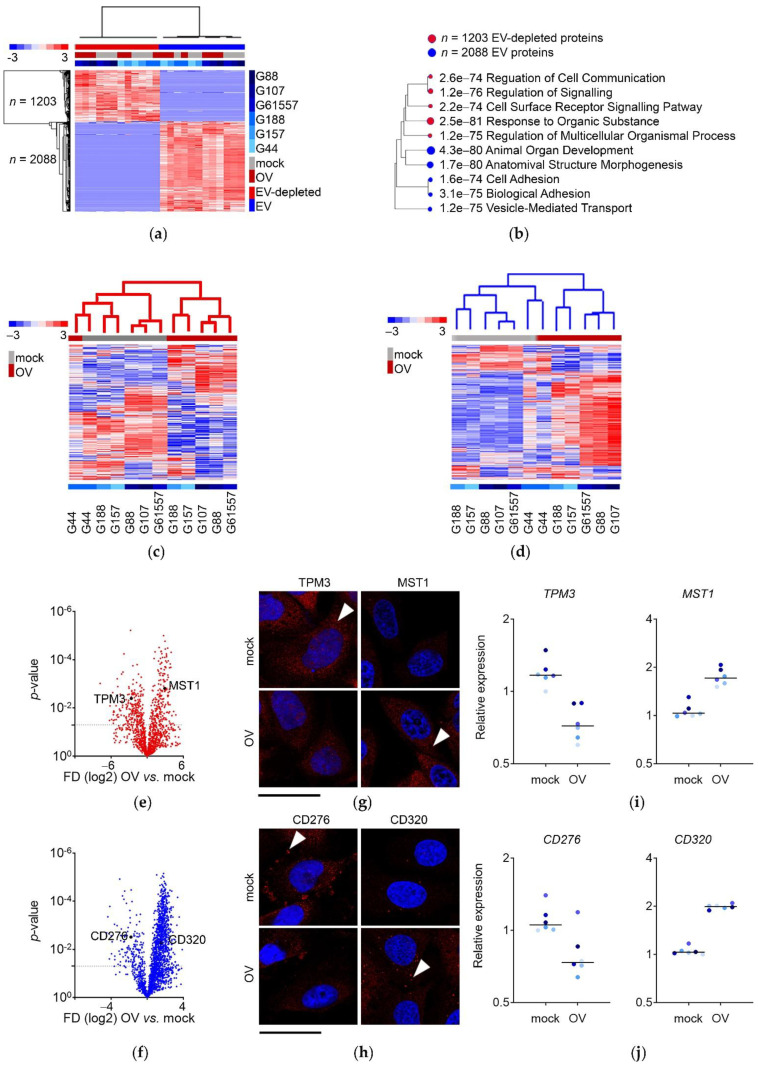
Proteins secreted by mock- and OV-infected GSCs ended up in non-overlapping EV-depleted and EV fractions. (**a**) The composition of EV-depleted and EV fractions of the secretome was distinctly dissimilar. Heatmap with unsupervised hierarchical clustering for GSCs mock or OV infected (*n* = 6 per variant) based on mass spectroscopy analysis in EV-depleted and EV fractions (*n* = 3291 proteins (1203 EV-depleted and 2088 EV)). (**b**) Proteins secreted EV-depleted and encapsulated in EV have different biological functions. A hierarchical clustering tree summarizes the biological functions of proteins identified by mass spectroscopy in EV-depleted and EV fractions, EV-depleted (red, *n* = 1203) and EV (blue, *n* = 2088). Bigger dots indicate more significant *p*-values. (**c**,**d**) OV infection shifts the protein composition of the EV-depleted and EV secretome. Heatmap with unsupervised hierarchical clustering for GSCs mock or OV infected (*n* = 6 per variant) based on mass spectroscopy analysis in EV-depleted (**c**, red tree, *n* = 1203) and EV fractions (**d**, blue tree, *n* = 2088). (**e**,**f**) GSC secretome is deregulated upon OV infection. Volcano plot for GSCs mock or OV infected (*n* = 6 per variant); dashed lines indicating *p*-value cutoff; based on mass spectroscopy analysis in EV-depleted (**e**, red, *n* = 1203) and EV fractions (**f**, blue, *n* = 2088). Proteins selected for validation are indicated. (**g**,**h**) The expression and localization of EV-depleted and EV proteins were affected by OV infection. Representative micrographs of selected proteins in mock or OV infected GSCs by fluorescent confocal microscopy; red arrows: antibody signal, blue: DAPI; scale bar: 25µm. (**i**,**j**) Deregulation of secretome upon OV infection is linked to the expression of mRNA transcripts. qPCR analysis of selected genes encoding for selected proteins in GSCs mock or OV infected (*n* = 6 per variant); scatter plot with mean value; EV-depleted (**i**), and EV (**j**) protein markers.

**Figure 4 cancers-13-01287-f004:**
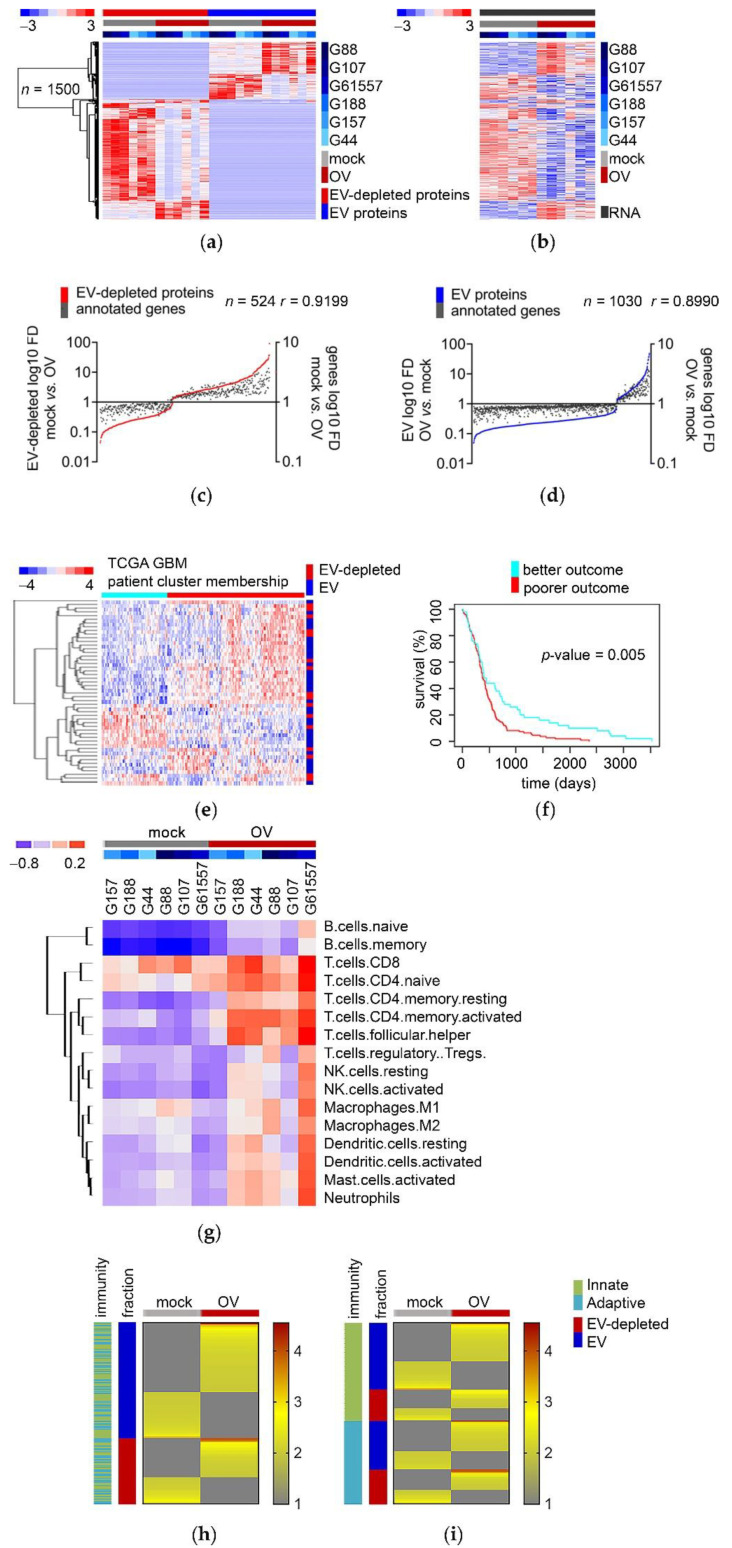
The re-configuration of GSCs’ transcriptome upon OV infection predicts the composition of their “immune mode” secretome. (**a**) OV infection profoundly affects the composition of the GSC secretome in both EV-depleted and EV fractions. Heatmap with supervised clustering for GSCs mock or OV infected (*n* = 6 per variant) based on mass spectroscopy analysis in EV-depleted and EV fractions (*n* = 1500 proteins, cutoffs between groups mock vs. OV in both fractions: *p*-value < 0.05, FD > 2). (**b**) OV infection shifts the transcriptome encoding for GSC secretome. Heatmap with unsupervised clustering for GSCs mock or OV infected (*n* = 6 per variant) based on mRNA transcripts (*n* = 1500 genes encoding for proteins shown on a panel a, based on Arraystar™ human mRNA array, *p*-value < 0.05). (**c**,**d**) The rearrangement of GSCs’ protein secretome is mostly mirrored by changes in their respective genes’ expression. Scatter plots of averaged association between GSC EV-depleted proteins (c, red) or EV protein (d, blue) and their annotated genes (gray) upon OV infection, based on a 524 or 1030-gene signature, respectively (*n* = 6 GSC per group, *p*-value < 0.05). e,f: Secretome proteins, encoded by genes predicting patients’ outcome, are found in both EV-depleted and EV fractions. The list of 1500 genes (see panel a) was filtered to the top-50 varied genes using TCGA data associated with glioblastoma. (**e**) Heatmap of genes with color annotations according to profile similarity stratification membership cluster (horizontal cluster) and EV-depleted and EV fractions (vertical cluster). (**f**) Kaplan–Meier curves show survival analysis stratified according to their cluster membership (see panel e). (**g**) GSCs boost the expression of immune response genes encoding for their secretome upon OV infection. Heatmap classifies GSCs mock, or OV infected (*n* = 6 per variant) to deconvolute gene expression profiles into cell-type-specific profiles based on gene expression (*n* = 1500) signatures. (**h**,**i**) Immune response secretome signature is affected in GSCs upon OV infection, regardless of secretome fractions and immunity type. Heatmaps classify GSCs mock or OV infected (*n* = 6 per variant) to deconvolute secretome profiles into EV-depleted and EV (**h**) and type of immunity (**i**) specific profiles, based on immune response protein signatures (*n* = 220); color bars indicate secretome fractions (dark blue-red) and type of immunity (light blue-green).

## Data Availability

The data presented in this study available are openly available in GEO Portal, accession number: GEO: GSE155247

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
