# Peer review of "Oncolytic Virus Therapy Alters the Secretome of Targeted Glioblastoma Cells"

_cancers, 2021, doi:10.3390/cancers13061287_

Round 1

Reviewer 1 Report

Godlewski and colleagues provide a comprehensive analysis of the influence of oncolytic virus infection on the secretome of human glioma stem cells. This study is of particular interest because it investigates the relations between oncolytic viruses, extracellular vesicles and the immunomodulation of the tumor microenvironment, three topics that are currently under the spotlight in cancer research.

However, I raise several points that will need to be addressed by the authors before publication. I feel that some conclusions are not completely supported by the data and that the overall objective of the study is not perfectly clear. The data are descriptive and will provide resources for the domain of oncolytic immunotherapy, but I think that some conclusions are overstated as it is not completely surprising that a viral infection will affect secretory processes and impact the production of immunity-related proteins.

Major revisions:

  • Lines 136-141: To purify EVs, the authors ultra-centrifugated the supernatants at 100,000g once but did not perform a second ultracentrifugation in PBS (or other washing technique) while it is recommended for studies on EVs to remove contaminating proteins (see ‘Minimal information for studies of extracellular vesicles MISEV 2018’). This could have an impact on the subsequent proteomic analyses and the authors should discuss this. Another important issue is that their EV purification strategy will end in co-purifying HSV particles (150-200 nm) since the 0,22µm filtration will not eliminate them. In the subsequent analyses this will deeply impact the detection of viral proteins that are reported as being in EVs while they can just be viral particles on their own. The authors should attempt to provide results showing the respective amounts of EVs and viral particles, maybe by titering the virus in their samples.
  • Line 208: The authors indicate that before proteomics analyses their samples were processed on a 30 kDa MWCO device. I am not perfectly clear on the reason for performing this preparatory step as it may result in losing proteins <30kDa for the subsequent analyses. Please explain.
  • Lines 362-365 & 377: The micrographs from Figure 2a give little information whether the cell membranes are actually intact. An alternative assay is needed (viability dye staining, release of intracellular molecules…). This would also complement Figure 2b as the increase of released proteins in the supernatant may be due to the lysis of at least a part of the infected cells. When looking at Table 2, some nuclear proteins are found in the proteomics analysis that may be markers of cell lysis. In addition, the proximity of EVs and viral particles is not clear from the pictures, a co-staining with an EV-specific marker such as CD63 would help substantiating this proximity.
  • Lines 369-370: The NTA analysis on Figure 2e actually shows some differences in the profiles of the different samples that could be discussed. Some events are also recorded over 220nm while the samples were filtered at 0,22µm. I wonder whether contamination of samples by viral particles may have an impact on these profiles. In addition, it is not clear from the Figure 2e whether only OV-infected samples are presented. Finally, Figure 2d is not sufficient to characterize the EVs; as recommended by the MISEV 2018, the authors should analyze (e.g. western-blot) the expression of positive (e.g. CD63, CD9, CD81) and negative markers of EVs to claim that they are bona fide
  • In Figure 2g, mock-treated HSV-sensitive cell lines show a certain amount of viral proteins, which is surprising. When looking at Table 2, the viral sequence library (cited in line 238) contains sequences from a variety of viruses that are not relevant for this article. I wonder why proteins from other viruses are found in the samples, whereas at the same time there is no mention of HSV proteins in the data. Would it be possible to reanalyze the proteomics data to search for HSV protein sequences? Please discuss.
  • Lines 401-402: The micrographs provided in Figure 3g-h are inconclusive in my opinion. I could not understand to what exactly the arrows are pointing, the magnification seems too low to even be able to point a particular event or structure. A co-staining with a protein specific for EVs such as CD63 would give more reliable information. Regarding the levels of expression for TPM3, MST1, CD276 and CD320, I think that Figure S3d is more conclusive and should replace Figures 3g-h. This should be complemented by an assay (e.g. western-blot) that will provide more robust results than a limited number of cells observed by microscopy. Also, I don’t really understand the relevance of Figure S3e in the manuscript.
  • Keys are missing to fully understand Table 2, for instance the names of the viruses from the viral library.

Minor revisions:

  • Overall I feel that the English language and style need to be improved, especially for the Simple Summary and the Abstract. As I am not a native English speaker, I however leave the Editor and the authors to decide whether there is a need for extensive editing.
  • Lines 21-23: there is no logical relation between the two parts of the sentence.
  • The Introduction is a little too long and may benefit from being written in a more scientific way.
  • Line 88: I could not find the Graphical Abstract in the provided material.
  • Line 104: please edit “FGF” to “FGF2”.
  • Line 146: purification of microRNAs is described here but I could not find any data related to this in the manuscript.
  • Lines 156-158: please indicate the antibody working concentrations instead of the dilutions.
  • Line 291: given the experiments reported in the manuscript, I believe that “unpaired” t-tests would be more appropriate.
  • Line 334-336: the conclusion for Figure S1g-h is not as clear as the authors state, considering that “metabolic functions” are also seen in the up-regulated genes in OV vs mock. Throughout the paper these pathway analyses are not very informative as they encompass very large cellular mechanisms which labels are sometimes quite obscure.
  • Line 348: “release” may be more appropriate than “secretion” as it is not formally demonstrated that this is an active process.
  • Lines 389-390 & 522-523: this difference of proteome between the two fractions is far from surprising. I would remove “remarkably” in the figure legend (line 431).
  • Lines 392-394 & 405-407: again, this is not completely surprising in my opinion.
  • Figure 4: panel letters are to be put in bold in the legend for smooth reading. Color legend for “patient cluster membership” in panel e is not defined.
  • As for the introduction, the discussion would benefit from a more scientific and precise writing.
  • Line 492: please give the reference of the “ongoing clinical trial”.
  • Ref 63 is cited twice at lines 495 and 518 but does not seem to be relevant for the respective sentences.
  • Lines 505-508: this seems quite out of the scope of the discussion and I would advise to remove it.
  • Lines 508-510: I don’t think that this concept is very new and deserves to be renamed as proposed by the authors.
  • The authors deposited their transcriptomic data on the GEO portal. I would advise them to deposit also their data in EV databases like EV-TRACK.

Author Response

REVIEWER #1

Godlewski and colleagues provide a comprehensive analysis of the influence of oncolytic virus infection on the secretome of human glioma stem cells. This study is of particular interest because it investigates the relations between oncolytic viruses, extracellular vesicles and the immunomodulation of the tumor microenvironment, three topics that are currently under the spotlight in cancer research.

However, I raise several points that will need to be addressed by the authors before publication. I feel that some conclusions are not completely supported by the data and that the overall objective of the study is not perfectly clear. The data are descriptive and will provide resources for the domain of oncolytic immunotherapy, but I think that some conclusions are overstated as it is not completely surprising that a viral infection will affect secretory processes and impact the production of immunity-related proteins.

We are thankful for the reviewer's comments. Addressing them will improve the manuscript message and will clarify details accurately pointed by the reviewer. Both manuscript and figures were revised to include requested datasets and to address reviewers’ critiques and comments.

Major revisions:

  • Lines 136-141: To purify EVs, the authors ultra-centrifugated the supernatants at 100,000g once but did not perform a second ultracentrifugation in PBS (or other washing technique) while it is recommended for studies on EVs to remove contaminating proteins (see ‘Minimal information for studies of extracellular vesicles MISEV 2018’). This could have an impact on the subsequent proteomic analyses and the authors should discuss this. Another important issue is that their EV purification strategy will end in co-purifying HSV particles (150-200 nm) since the 0,22µm filtration will not eliminate them. In the subsequent analyses this will deeply impact the detection of viral proteins that are reported as being in EVs while they can just be viral particles on their own. The authors should attempt to provide results showing the respective amounts of EVs and viral particles, maybe by titering the virus in their samples.

We agree with the reviewer's comments and we added/corrected details of the methodology, to explain the inaccuracy pointed by the reviewer. The changes include new flowchart depicting used methodology (new Figure S2a), and more details were added to the Materials and Methods section.

  • Line 208: The authors indicate that before proteomics analyses their samples were processed on a 30 kDa MWCO device. I am not perfectly clear on the reason for performing this preparatory step as it may result in losing proteins <30kDa for the subsequent analyses. Please explain.

We agree that this is an incorrect statement. MWCO (3 kDa not 30 kDa) was used to increase the concentration of protein for electron microscopy analysis of EV-depleted fraction. The change is reflected in Methodology section.

  • Lines 362-365 & 377: The micrographs from Figure 2a give little information whether the cell membranes are actually intact. An alternative assay is needed (viability dye staining, release of intracellular molecules). This would also complement Figure 2b as the increase of released proteins in the supernatant may be due to the lysis of at least a part of the infected cells. When looking at Table 2, some nuclear proteins are found in the proteomics analysis that may be markers of cell lysis. In addition, the proximity of EVs and viral particles is not clear from the pictures, a co-staining with an EV-specific marker such as CD63 would help substantiating this proximity.

Indeed, predominantly nuclear RNA-, and DNA-binding proteins were found in EVs, yet in our opinion, if the release of these proteins would be due to the lysis of cells, we would detect them in EV-depleted fraction. So, to address reviewer’s concern, we provided the requested CD63 marker analysis in the revised version (new Figure 2b-c).

  • Lines 369-370: The NTA analysis on Figure 2e actually shows some differences in the profiles of the different samples that could be discussed. Some events are also recorded over 220nm while the samples were filtered at 0,22µm. I wonder whether contamination of samples by viral particles may have an impact on these profiles. In addition, it is not clear from the Figure 2e whether only OV-infected samples are presented. Finally, Figure 2d is not sufficient to characterize the EVs; as recommended by the MISEV 2018, the authors should analyze (e.g. western-blot) the expression of positive (e.g. CD63, CD9, CD81) and negative markers of EVs to claim that they are bona fide

We agree that the presence of viral particles may affect the profile, so we provided the requested analysis, and incorporated it into new Figure 2g, and we added EV and EV-depleted fraction markers analysis (new Figure S2e).

  • In Figure 2g, mock-treated HSV-sensitive cell lines show a certain amount of viral proteins, which is surprising. When looking at Table 2, the viral sequence library (cited in line 238) contains sequences from a variety of viruses that are not relevant for this article. I wonder why proteins from other viruses are found in the samples, whereas at the same time there is no mention of HSV proteins in the data. Would it be possible to reanalyze the proteomics data to search for HSV protein sequences? Please discuss.

Indeed we found proteins from distinct virus species, including herpes simplex virus in EV (denoted as a human herpes virus). As viral protein species in the library nomenclature in Table 2 was misleading, we included full Latin name of species.

But detection of viral proteins pointed by reviewer is very interesting. To discuss this, we included the following paragraph in the Discussion: “The origin of glioblastoma as a result of virus infection was proposed, and especially promising data was found for cytomegalovirus /reference/. But low cellular detection born some skepticism in the community. While not directly pointed by current analysis, it will be interesting to find whether the deposition of cytomegalovirus proteins is detectable only in glioblastoma-derived EVs. It would support our observation that while viral microRNAs are not detectable in glioblastoma cells, we found them in EVs /reference/. The explanation can be as trivial as a different ratio of viral vs. cell host proteome, yet the overarching explanation is still lacking and will be the subject of follow-up studies”.

  • Lines 401-402: The micrographs provided in Figure 3g-h are inconclusive in my opinion. I could not understand to what exactly the arrows are pointing, the magnification seems too low to even be able to point a particular event or structure. A co-staining with a protein specific for EVs such as CD63 would give more reliable information. Regarding the levels of expression for TPM3, MST1, CD276 and CD320, I think that Figure S3d is more conclusive and should replace Figures 3g-h. This should be complemented by an assay (e.g. western-blot) that will provide more robust results than a limited number of cells observed by microscopy. Also, I don’t really understand the relevance of Figure S3e in the manuscript.

To address reviewer’s concern, we included immunostaining of markers with DAPI co-staining (Figure 3g-h). The fraction marker analysis was also provided, as explained above (Figure 2b-c, Figure S2d). Finally, we included a sentence providing rationale for the analysis of sub-cellular localization of markers in question in cancer cells other than glioblastoma (Figure S3e).

  • Keys are missing to fully understand Table 2, for instance the names of the viruses from the viral library.

Table 2 was revised according to the reviewer’s suggestions.

Minor revisions:

  • Overall I feel that the English language and style need to be improved, especially for the Simple Summary and the Abstract. As I am not a native English speaker, I however leave the Editor and the authors to decide whether there is a need for extensive editing.

As per reviewer’s request, Simple Summary and Abstract were lightly edited.

  • Lines 21-23: there is no logical relation between the two parts of the sentence.

The sentence was corrected.

  • The Introduction is a little too long and may benefit from being written in a more scientific way.

As per reviewer’s request, the Introduction was shortened and edited.

  • Line 88: I could not find the Graphical Abstract in the provided material.

As Graphical Abstract is not available at this stage, we removed the above-mentioned reference.

  • Line 104: please edit “FGF” to “FGF2”.

Corrected.

  • Line 146: purification of microRNAs is described here but I could not find any data related to this in the manuscript.

Corrected

  • Lines 156-158: please indicate the antibody working concentrations instead of the dilutions.

Done

  • xLine 291: given the experiments reported in the manuscript, I believe that “unpaired” t-tests would be more appropriate

The statistical methodology was described for both biological and technical replicates.

  • Line 334-336: the conclusion for Figure S1g-h is not as clear as the authors state, considering that “metabolic functions” are also seen in the up-regulated genes in OV vs mock. Throughout the paper these pathway analyses are not very informative as they encompass very large cellular mechanisms which labels are sometimes quite obscure.

We clarified this by specifying simultaneous up-regulation of RNA metabolism (suggestive of altered gene expression) and down-regulation of housekeeping metabolism.

  • Line 348: “release” may be more appropriate than “secretion” as it is not formally demonstrated that this is an active process.

Revised as per reviewer’s suggestion.

  • Lines 389-390 & 522-523: this difference of proteome between the two fractions is far from surprising. I would remove “remarkably” in the figure legend (line 431). Lines 392-394 & 405-407: again, this is not completely surprising in my opinion.

Revised as per reviewer’s suggestion.

  • Figure 4: panel letters are to be put in bold in the legend for smooth reading. Color legend for “patient cluster membership” in panel e is not defined.

Patient cluster membership is common for panels e and f. The legend has been revised.

  • As for the introduction, the discussion would benefit from a more scientific and precise writing.

As per reviewer’s request, the Discussion was revised.

  • Line 492: please give the reference of the “ongoing clinical trial”.

References were added.

  • Ref 63 is cited twice at lines 495 and 518 but does not seem to be relevant for the respective sentences.

Reference was removed from irrelevant statement.

  • Lines 505-508: this seems quite out of the scope of the discussion and I would advise to remove it. Lines 508-510: I don’t think that this concept is very new and deserves to be renamed as proposed by the authors.

The paragraph was removed as per reviewer’s request.

  • The authors deposited their transcriptomic data on the GEO portal. I would advise them to deposit also their data in EV databases like EV-TRACK.

As EV-TRACK allows the deposition of up to 10 samples, and we have 12 samples, we will deposit data in two sets.

Reviewer 2 Report

This is an interesting paper describing changes in GBM cell transcriptomes and proteomes upon introduction of oncolytic virus and how these changes may affect the secretome and in turn the very important immune-responses. The datasets are useful for researchers in the field. I think that the authors could have been a bit more detailed in the description of the results in the abstract and in the paper bulk. Also the hypothesis, the idea of the study, why this is important is not that clearly stated. But overall, it is a very well performed study that contains a lot of useful data for the glioma researcher in general.

Author Response

REVIEWER #2

This is an interesting paper describing changes in GBM cell transcriptomes and proteomes upon introduction of oncolytic virus and how these changes may affect the secretome and in turn the very important immune-responses. The datasets are useful for researchers in the field. I think that the authors could have been a bit more detailed in the description of the results in the abstract and in the paper bulk. Also the hypothesis, the idea of the study, why this is important is not that clearly stated. But overall, it is a very well performed study that contains a lot of useful data for the glioma researcher in general.

We are thankful for reviewer’s appreciation. We provided the rationale for our study in the revised version of the manuscript. Introduction and Discussion were edited to address reviewer’s comments. Also we provided both cartoon with the flow chart of experimental design (FigS.2a) and new datasets to clarify and strengthen the message of our research (Fig 2b-c), (FS2e).

Reviewer 3 Report

In this manuscript, the authors used transcriptomics and proteomics analysis of 6 different glioblastoma tumor cells infected with oncolytic herpes virus. They concentrate their efforts on the secretome, an original approach in this context.

Overall this is quite an extensive analysis and data are clear and could be a useful addition to the literature on oncolytic viruses. However, some precision should be added to clarify the interpretation of these data, as suggested below.

1) More details should be given on the definition of EV and EV-depleted fractions. Is there a reference for this definition. It does seem crude to consider a pellet and supernatant without additional purification. What is looked at exactly is not that clear to me. 

2) Is there viral particles and are those pelleted or not under the conditions used? Is it known if the herpes virus could bind to cellular proteins or harbor cellular proteins specifically or non-specifically incorporated to the virions. An electron microscopy analysis of both fractions will be interesting.

3) For the EV fraction, the addition of immuno-electron microscopy to directly examine some of the enriched protein and to confirm their actual incorporation to EV will further strengthen the conclusions and ruled out that these are virion-associated.

Author Response

REVIEWER #3

In this manuscript, the authors used transcriptomics and proteomics analysis of 6 different glioblastoma tumor cells infected with oncolytic herpes virus. They concentrate their efforts on the secretome, an original approach in this context.

Overall this is quite an extensive analysis and data are clear and could be a useful addition to the literature on oncolytic viruses. However, some precision should be added to clarify the interpretation of these data, as suggested below.

  • More details should be given on the definition of EV and EV-depleted fractions. Is there a reference for this definition. It does seem crude to consider a pellet and supernatant without additional purification. What is looked at exactly is not that clear to me. 

The terms EV and EV-depleted fractions have been introduced by us for the purpose of this study. We agree that the methodology requires better description. To this end we clarified the Methods section, provided a flow chart of experimental procedures (new Figure S2a), and included new datasets (new Figure S2e).

  • Is there viral particles and are those pelleted or not under the conditions used? Is it known if the herpes virus could bind to cellular proteins or harbor cellular proteins specifically or non-specifically incorporated to the virions. An electron microscopy analysis of both fractions will be interesting.

The viral particles are within the EV pellets as underlined by the detection of viral proteins exclusively in EV but not EV-depleted fraction (new Table S2). We also provided an electron microscopy analysis of both fractions as requested (new Figure 2b).

  • For the EV fraction, the addition of immuno-electron microscopy to directly examine some of the enriched protein and to confirm their actual incorporation to EV will further strengthen the conclusions and ruled out that these are virion-associated.

We provided an electron microscopy with immune-labeling of CD63 marker in both fractions (new Figure 2b), and the analysis of peptides for fraction markers (new Figure S2e). 

Round 2

Reviewer 1 Report

Godlewski and colleagues provide a comprehensive analysis of the influence of oncolytic virus infection on the secretome of human glioma stem cells. This study is of particular interest because it investigates the relations between oncolytic viruses, extracellular vesicles and the immunomodulation of the tumor microenvironment, three topics that are currently under the spotlight in cancer research.

However, I raise several points that will need to be addressed by the authors before publication. I feel that some conclusions are not completely supported by the data and that the overall objective of the study is not perfectly clear. The data are descriptive and will provide resources for the domain of oncolytic immunotherapy, but I think that some conclusions are overstated as it is not completely surprising that a viral infection will affect secretory processes and impact the production of immunity-related proteins.

We are thankful for the reviewer's comments. Addressing them will improve the manuscript message and will clarify details accurately pointed by the reviewer. Both manuscript and figures were revised to include requested datasets and to address reviewers’ critiques and comments.

Major revisions:

  • Lines 136-141: To purify EVs, the authors ultra-centrifugated the supernatants at 100,000g once but did not perform a second ultracentrifugation in PBS (or other washing technique) while it is recommended for studies on EVs to remove contaminating proteins (see ‘Minimal information for studies of extracellular vesicles MISEV 2018’). This could have an impact on the subsequent proteomic analyses and the authors should discuss this. Another important issue is that their EV purification strategy will end in co-purifying HSV particles (150-200 nm) since the 0,22µm filtration will not eliminate them. In the subsequent analyses this will deeply impact the detection of viral proteins that are reported as being in EVs while they can just be viral particles on their own. The authors should attempt to provide results showing the respective amounts of EVs and viral particles, maybe by titering the virus in their samples.

We agree with the reviewer's comments and we added/corrected details of the methodology, to explain the inaccuracy pointed by the reviewer. The changes include new flowchart depicting used methodology (new Figure S2a), and more details were added to the Materials and Methods section.

I thank the authors for providing new methodology details for the purification of EVs. However, the second point was not discussed. I do not request additional experiments but I think it should be written in the main text (and Figure S2a) that the ‘EV fraction’ is actually a ‘EV + HSV fraction’. This can deeply impact the subsequent proteomic analyses, for instance by ‘diluting’ the cellular proteins among viral proteins. Please discuss.

  • Line 208: The authors indicate that before proteomics analyses their samples were processed on a 30 kDa MWCO device. I am not perfectly clear on the reason for performing this preparatory step as it may result in losing proteins <30kDa for the subsequent analyses. Please explain.

We agree that this is an incorrect statement. MWCO (3 kDa not 30 kDa) was used to increase the concentration of protein for electron microscopy analysis of EV-depleted fraction. The change is reflected in Methodology section.

Thank you for this clarification.

  • Lines 362-365 & 377: The micrographs from Figure 2a give little information whether the cell membranes are actually intact. An alternative assay is needed (viability dye staining, release of intracellular molecules). This would also complement Figure 2b as the increase of released proteins in the supernatant may be due to the lysis of at least a part of the infected cells. When looking at Table 2, some nuclear proteins are found in the proteomics analysis that may be markers of cell lysis. In addition, the proximity of EVs and viral particles is not clear from the pictures, a co-staining with an EV-specific marker such as CD63 would help substantiating this proximity.

Indeed, predominantly nuclear RNA-, and DNA-binding proteins were found in EVs, yet in our opinion, if the release of these proteins would be due to the lysis of cells, we would detect them in EV-depleted fraction. So, to address reviewer’s concern, we provided the requested CD63 marker analysis in the revised version (new Figure 2b-c).

I agree with the authors regarding their explanation for minimal cell lysis demonstrated by the fact that only few intracellular proteins are found in the EV-depleted fraction. The new Figure 2b-c is a great addition to the paper by characterizing the EV fraction with the CD63 marker. However, this does not support the claim that HSV colocalizes with EVs. Palm-tdTomato is a marker for membranes, including the plasma membrane, and cannot be used as an EV marker. A specific staining (and a better magnification) is needed for such a statement. I strongly advise that the authors tone down their affirmation (lines 363-364), especially considering that it is not of particular importance for the interpretation of their subsequent results.

Line 369, please edit 'CD36' into 'CD63'.

  • Lines 369-370: The NTA analysis on Figure 2e actually shows some differences in the profiles of the different samples that could be discussed. Some events are also recorded over 220nm while the samples were filtered at 0,22µm. I wonder whether contamination of samples by viral particles may have an impact on these profiles. In addition, it is not clear from the Figure 2e whether only OV-infected samples are presented. Finally, Figure 2d is not sufficient to characterize the EVs; as recommended by the MISEV 2018, the authors should analyze (e.g. western-blot) the expression of positive (e.g. CD63, CD9, CD81) and negative markers of EVs to claim that they are bona fide

We agree that the presence of viral particles may affect the profile, so we provided the requested analysis, and incorporated it into new Figure 2g, and we added EV and EV-depleted fraction markers analysis (new Figure S2e).

I thank the authors for the new Figure 2g that clearly shows the differences in profiles between EVs from either control or OV-infected cells. The presence of particles over 220nm despite the 0.22µm filtration step may be discussed in the results. Is this a proof of the expected “contamination” with viral particles? I also greatly appreciate the new Figure S2g for EV characterization.

  • In Figure 2g, mock-treated HSV-sensitive cell lines show a certain amount of viral proteins, which is surprising. When looking at Table 2, the viral sequence library (cited in line 238) contains sequences from a variety of viruses that are not relevant for this article. I wonder why proteins from other viruses are found in the samples, whereas at the same time there is no mention of HSV proteins in the data. Would it be possible to reanalyze the proteomics data to search for HSV protein sequences? Please discuss.

Indeed we found proteins from distinct virus species, including herpes simplex virus in EV (denoted as a human herpes virus). As viral protein species in the library nomenclature in Table 2 was misleading, we included full Latin name of species.

But detection of viral proteins pointed by reviewer is very interesting. To discuss this, we included the following paragraph in the Discussion: “The origin of glioblastoma as a result of virus infection was proposed, and especially promising data was found for cytomegalovirus /reference/. But low cellular detection born some skepticism in the community. While not directly pointed by current analysis, it will be interesting to find whether the deposition of cytomegalovirus proteins is detectable only in glioblastoma-derived EVs. It would support our observation that while viral microRNAs are not detectable in glioblastoma cells, we found them in EVs /reference/. The explanation can be as trivial as a different ratio of viral vs. cell host proteome, yet the overarching explanation is still lacking and will be the subject of follow-up studies”.

I apologize but I could not find the new version of Table S2 with the full Latin names of viruses to reanalyze the data. I am not sure whether this is an issue with the submission/reviewing website.

Even if the discussion about the relation between CMV and glioblastoma is of some interest, it is still hard to understand the abundance of so many viral proteins (from cowpox, vaccinia…) in the EV fractions. When looking at the data for G44, it does not make a lot of sense. I strongly advise for the authors to verify their data for this table, in particular regarding the viral library results. Would it be possible that some HSV proteins are identified by spectrometry but attributed to other types of viruses?

  • Lines 401-402: The micrographs provided in Figure 3g-h are inconclusive in my opinion. I could not understand to what exactly the arrows are pointing, the magnification seems too low to even be able to point a particular event or structure. A co-staining with a protein specific for EVs such as CD63 would give more reliable information. Regarding the levels of expression for TPM3, MST1, CD276 and CD320, I think that Figure S3d is more conclusive and should replace Figures 3g-h. This should be complemented by an assay (e.g. western-blot) that will provide more robust results than a limited number of cells observed by microscopy. Also, I don’t really understand the relevance of Figure S3e in the manuscript.

To address reviewer’s concern, we included immunostaining of markers with DAPI co-staining (Figure 3g-h). The fraction marker analysis was also provided, as explained above (Figure 2b-c, Figure S2d). Finally, we included a sentence providing rationale for the analysis of sub-cellular localization of markers in question in cancer cells other than glioblastoma (Figure S3e).

The DAPI co-staining in Figure 3g-h helps understanding the localization of the studied proteins but I would advise to remove the arrows that give no specific information. The explanation added for Figure S3e is helpful.

  • Keys are missing to fully understand Table 2, for instance the names of the viruses from the viral library.

Table 2 was revised according to the reviewer’s suggestions.

As explained above, I could not have access to the updated version of Table S2.

Minor revisions:

  • Overall I feel that the English language and style need to be improved, especially for the Simple Summary and the Abstract. As I am not a native English speaker, I however leave the Editor and the authors to decide whether there is a need for extensive editing.

As per reviewer’s request, Simple Summary and Abstract were lightly edited.

Noted.

  • Lines 21-23: there is no logical relation between the two parts of the sentence.

The sentence was corrected.

Noted.

  • The Introduction is a little too long and may benefit from being written in a more scientific way.

As per reviewer’s request, the Introduction was shortened and edited.

Noted.

  • Line 88: I could not find the Graphical Abstract in the provided material.

As Graphical Abstract is not available at this stage, we removed the above-mentioned reference.

Noted.

  • Line 104: please edit “FGF” to “FGF2”.

Corrected.

Noted.

  • Line 146: purification of microRNAs is described here but I could not find any data related to this in the manuscript.

Corrected.

Noted.

  • Lines 156-158: please indicate the antibody working concentrations instead of the dilutions.

Done.

Noted.

  • Line 291: given the experiments reported in the manuscript, I believe that “unpaired” t-tests would be more appropriate

The statistical methodology was described for both biological and technical replicates.

Noted.

  • Line 334-336: the conclusion for Figure S1g-h is not as clear as the authors state, considering that “metabolic functions” are also seen in the up-regulated genes in OV vs mock. Throughout the paper these pathway analyses are not very informative as they encompass very large cellular mechanisms which labels are sometimes quite obscure.

We clarified this by specifying simultaneous up-regulation of RNA metabolism (suggestive of altered gene expression) and down-regulation of housekeeping metabolism.

Noted.

  • Line 348: “release” may be more appropriate than “secretion” as it is not formally demonstrated that this is an active process.

Revised as per reviewer’s suggestion.

Noted.

  • Lines 389-390 & 522-523: this difference of proteome between the two fractions is far from surprising. I would remove “remarkably” in the figure legend (line 431). Lines 392-394 & 405-407: again, this is not completely surprising in my opinion.

Revised as per reviewer’s suggestion.

Noted.

  • Figure 4: panel letters are to be put in bold in the legend for smooth reading. Color legend for “patient cluster membership” in panel e is not defined.

Patient cluster membership is common for panels e and f. The legend has been revised.

Noted.

  • As for the introduction, the discussion would benefit from a more scientific and precise writing.

As per reviewer’s request, the Discussion was revised.

Noted.

  • Line 492: please give the reference of the “ongoing clinical trial”.

References were added.

Noted.

  • Ref 63 is cited twice at lines 495 and 518 but does not seem to be relevant for the respective sentences.

Reference was removed from irrelevant statement.

Noted.

  • Lines 505-508: this seems quite out of the scope of the discussion and I would advise to remove it. Lines 508-510: I don’t think that this concept is very new and deserves to be renamed as proposed by the authors.

The paragraph was removed as per reviewer’s request.

Noted.

  • The authors deposited their transcriptomic data on the GEO portal. I would advise them to deposit also their data in EV databases like EV-TRACK.

As EV-TRACK allows the deposition of up to 10 samples, and we have 12 samples, we will deposit data in two sets.

I was not aware of this limitation but I thank the authors for their efforts in depositing their results in this database.

Reviewer 3 Report

The authors have reviewed their manuscript following reviewers suggestions. Overall the manuscript has improved and present an interesting story.

Author Response

Thank you, we appreciate your input.